# Nonlinear relationship between money market rate and stock market liquidity in China: A multifractal analysis

**Yihong Sun, Xuemei Yuan***

Institute of Finance, Shandong Collaborative Innovation Center for Capital Market Innovation and Development, University of Jinan, Jinan, 250002, China

* se_yuanxm@ujn.edu.cn

**Data Availability Statement:** The data are owned by a third party and authors do not have permission to share the data. The daily closing price and turnover rate of the Shanghai Composite Index are used to calculate the Shanghai Stock Market Liquidity (SHSML), and they are obtained from Wind. The data for SHIBOR (O/N) are collected from and freely available at http://www.shibor.org. Interested researchers can obtain the data of the closing price and turnover rate of Shanghai Composite Index if they buy the Win.d financial terminal and have the access to Win.d Database. Interested researchers can contact with

## Abstract

This paper employs the multifractal detrended cross-correlation analysis (MF-DCCA) model to estimate the nonlinear relationship between the money market rate and stock market liquidity in China from a multifractal perspective, leading to a better understanding of the complexity in the relationship between the interest rate and stock market liquidity. The empirical results show that the cross-correlations between the money market rate and stock market liquidity present antipersistence in the long run and that they tend to be positively persistent in the short run. The negative cross-correlations between the interest rate and stock market liquidity are more significant than the positive cross-correlations. Furthermore, the cross-correlations between the money market rate and stock market liquidity display multifractal characteristics, explaining the variations in the relationship between the interest rate and stock market liquidity at different time scales. In addition, the lower degree of multifractality in the cross-correlations between the money market rate and stock market liquidity confirms that it is effective for the interest rate to control stock market liquidity. The Chinese stock market liquidity is more sensitive to fluctuations in the money market rate in the short term and is inelastic in response to the money market rate in the long term. In particular, the positive cross-correlations between the money market rate and stock market liquidity in the short run become strong in periods of crises and emergencies. All the evidence proves that the interest rate policy is an emergency response rather than an effective response to mounting concerns regarding the economic impact of unexpected exogenous emergencies and that the interest rate cut policy will not be as effective as expected.

## Introduction

Against the background of the continuous escalation of trade friction, the large-scale outbreak of COVID-19, and the gradual rise of geopolitical risks, there are more uncertainties in economic conditions, making financial markets increasingly volatile and exposed to large liquidity risks. Monetary policy is the source of liquidity supply for the stock market, playing an important role in controlling stock market liquidity [1–3]. Monetary policy in terms of the interest

the following e-mail: sales@wind.com.cn or visit
the homepage of http://www.wind.com.cn to send
the data access requests. The authors had special
privileges in accessing the data from the Win.d
financial terminal through the Institute of Finance's
purchase of the terminal.

**Funding:** Xuemei Yuan 17BJY184 The National
Social Science Fund of China http://www.nopss.
gov.cn/ YES - decision to publish.

**Competing interests:** NO authors have competing
interests.

rate is one of the most common tools used by major central banks in the liquidity supervision of various countries because of the advantages of interest rates in terms of measurability, controllability, and relevance to financial markets. Interest rate adjustments are becoming more important in adding additional stock market liquidity [4,5]. However, some economists remain divided over whether to cut interest rates during periods of stress and uncertainties in the economy. For example, the new round of unlimited quantitative easing (QE), an effort to stop stock market liquidity slipping by the federal government amid COVID-19, has been questioned in both academic and business circles. What matters is to make clear the relationship between interest rate and stock market liquidity for social sustainability. In particular, the internal influence mechanism of interest rate on stock market liquidity is too complex to reach an agreement on the relationship between the interest rate and stock market liquidity in recent studies.

The interest rate policy can create different macroeconomic environments for favorable economic outcomes to meet specific needs, that is, central bank officials traditionally raise interest rates when the economy is growing as a way to combat inflation and tend to cut interest rates when they are guarding against a downturn as a way to try to encourage more borrowing and spending in the economy. As most studies have documented, there is a significant negative relationship between interest rates and stock market liquidity. For example, Marie-Hélène and Céline (2013) [6] provide evidence that a short-term interest rate cut caused a temporary decrease in volatility and an increase in liquidity in North American stock markets. This scenario is also validated by Jiang (2014) [7], who finds that an increase in the interest rate lowers stock liquidity since it points to a contractionary monetary policy and is likely to lead to a decline in providers' funding liquidity. This effect is larger for stocks with low market capitalization and low liquidity. Chu (2015) [8] investigates the dependence structure between the monetary policy and stock market liquidity, showing that a stock market with high liquidity is dependent on an expansionary monetary policy, as a low interest rate reduces constraints for margin borrowing and thus improves funding liquidity. Lawal et al. (2018) [9] employ the ARDL model and EGARCH model to confirm that interest rate cuts can directly reduce the cost of capital to change the present value of future cash flow and indirectly influence investment through credit channels to increase stock market liquidity. Zhang et al. (2019) [10] empirically study the relationship between monetary policy adjustment and stock liquidity from the impacts of corporate investment on stock liquidity and the underlying influencing mechanism, pointing out that the interest rate policy has a negative effect on stock market liquidity affected by corporate investment.

In contrast to the traditional view that there is a negative relationship between the interest rate and stock market liquidity, some studies obtain inconsistent findings suggesting that the relationship between the interest rate and stock market liquidity is positive during some special periods. They found a direction variation in the relationship between the interest rate and stock market liquidity. Gali and Gambetti (2015) [11] estimate the response of stock prices to monetary policy shocks using a time-varying coefficients VAR. They observe a rather counterintuitive result: a positive monetary policy shock actually has a positive impact on the stock market, and it is unlikely to be accounted for by an endogenous response of the equity premium to the monetary policy shock. As Smimou and Khallouli (2015) [12] documented, the interest rate had a positive effect on stock market liquidity when the Euro was introduced in Eurozone countries. The explanation for the positive relationship between the interest rate and stock market liquidity may be that a component of economic growth exists in stock market liquidity. Similarly, Balafas et al. (2018) [13] present evidence that the negative relationship between the interest rate and UK stock market liquidity reversed its sign and tended to be significantly positive in financial crises. Li et al. (2019) [14] detect the relationship between a low

interest rate policy and the stock market by using the TVP-VAR model from theoretical and empirical perspectives, confirming the uncertain impact of the interest rate on stock market liquidity. They argue that interest rate cuts usually promote stock liquidity, but the opposite conditions are found in some periods.

The findings of recent studies also show a cyclical variation in the relationship between interest rates and stock market liquidity, with the response of stock market liquidity to interest rates being much stronger in different time and space scenarios. Arabinda and Alexander (2008) [15] find a much stronger response of the stock market to unexpected changes in the federal funds target rate in a recession and in tight credit market conditions, indicating the cyclical variation in the relationship between the interest rate and stock market liquidity. Similar results from Reinder et al. (2016) [16], Caraiani and Călin (2018) [17], and Collingro and Frenkel (2019) [18] also indicate some significant differences in the impact of the interest rate on stock market liquidity in the Euro area when looking at the results during and in the aftermath of a crisis by using an event study method, the time-varying Bayesian VAR model and the GMM approach. Their results suggest that the interest rate policy shows a stronger effect on stock market liquidity during stock market crises and financial crises, and the response of value and past loser stocks to interest rate changes is larger.

Previous studies have presented substantial evidence on the real relationship between the interest rate and stock market liquidity; however, the relationship between the interest rate and stock market liquidity is still uncertain. There are direction and cyclical variations in the relationship between them in different temporal scales and spatial scenarios. These variations may be associated with the complexity of the financial market. Since financial markets are not fully efficient markets, the time series of financial markets is autocorrelated and displays a fat-tail distribution rather than a Gaussian distribution. It is difficult for a general mathematical model to detect the real relationship between the nonstationary time series of interest rates and stock market liquidity. Therefore, there are some misunderstandings regarding the relationship between the interest rate and stock market liquidity under the framework of an efficient market hypothesis, causing the inconsistent findings obtained by previous studies.

The complex system has features similar to physical mechanics, such as fractal and chaotic structures. Multifractal theory, a subfield of dimension theory in dynamical systems, divides complex fractals into many small areas with different degrees of singularity to describe the laws contained in irregular complex phenomena in more detail, providing a new perspective to study the nonlinear correlations between complex systems. Based on multifractal theory, Hurst (1951) [19,20] first developed rescaled range analysis (R/S) for the study of long-memory time series. Peng et al. (1994) [21] proposed detrended fluctuation analysis (DFA) to address the issues of sensitivity to short-range autocorrelations in the R/S method. Kantelhardt et al. (2002) [22] advanced multifractal detrended fluctuation analysis (MF-DFA) to explore the multifractal characteristics of nonstationary time series. Then, Podobnik and Stanley (2008) [23] further developed detrended cross-correlation analysis (DCCA) to study the long-range cross-correlations between nonstationary time series based on the MF-DFA method. To study the multifractality of autocorrelated nonstationary time series, Zhou (2008) [24] introduced multifractal detrended cross-correlation analysis (MF-DCCA) by integrating DCCA into MF-DFA.

In addition, there are a number of techniques and approaches related to MF-DCCA to detect the long-term dependence and quantify cross-correlations between nonstationary time series. Zebende (2011) [25] proposed a new coefficient that is defined in terms of the DFA method and the DCCA method with the objective of quantifying the level of cross-correlation between nonstationary time series. Podobnik (2011) [26] used DCCA and proposed the detrended cross-correlation coefficient ($\rho DCCA(T, n)$) to quantify the cross-covariance and

the cross-correlation of two nonstationary time series. Jiang and Zhou (2011) [27] developed a class of MF-DCCA algorithms based on detrending moving-average analysis, called MF-XDMA. Wang et al. (2013) [28] used the empirical cross-correlation matrices constructed by the DCCA coefficient to show properties at different time scales in the US stock market. Yuan et al. (2015) [29] proposed detrended partial cross-correlation analysis (DPCCA) based on DCCA to quantify the relations of two nonstationary signals (with the influences of other signals removed) on different time scales. Then, Lin et al. (2018) [30] employed the detrended cross-correlation analysis (DCCA) coefficient and the detrended partial cross-correlation analysis (DPCCA) coefficient to investigate cross-correlations and net cross-correlations among five major world gold markets. Qian et al. (2015) [31] analyzed multifractal binomial measures masked with strong white noise and found that the MF-DPXA method quantifies the hidden multifractal nature.

More recently, the MF-DCCA method has become widely used for earthquake [32] and hydrology [33] studies, especially in relation to financial markets [34,35]. Zhang et al. (2018) [36] and Xiong et al. (2019) [37] introduced the MF-DCCA method to study the Chinese stock market. Zhang et al. (2018) [38] and Zhang et al. (2018) [39] adopted the MF-DCCA method to test the efficiency of the cryptocurrency market and investigated the cross-correlations of the return-volume relationship of the cryptocurrency market. Therefore, this paper employs the multifractal detrended cross-correlation analysis (MF-DCCA) model to examine the relationship between money market rate and stock market liquidity in China from a multifractal perspective. We find that the correlations between the money market rate and stock market liquidity are nonlinear and dynamic. Furthermore, the correlations between the money market rate and stock market liquidity in China also present multifractal characteristics, explaining the variations in the relationship between the interest rate and stock market liquidity.

The rest of this paper is organized as follows: section 2 introduces the approach we used in this study. Section 3 describes the data. Section 4 presents the empirical analysis. Section 5 offers discussions.

## Methodology

Take time series $x(i)$ and $y(i)$, $i = (1, 2, . . ., N)$, where N is the length of the time series. The MF-DCCA model includes seven steps:

Step 1: Calculate the cumulative distribution of the two time series.

$$X(i) = \sum_{k=1}^{i}(x(k) - \bar{x})$$

$$Y(i) = \sum_{k=1}^{i}(y(k) - \bar{y})$$

(1)

where $\bar{x}$ and $\bar{y}$ are the average values of time series $x(i)$ and $y(i)$, respectively.

Step 2: Divide the two sequences into nonoverlapping segments $s$ with equal length $N_S = int$ $[N/s]$.

Since the length $N$ of the time series is not always an integer multiple of the time scale $s$, a small part of the series $x(i)$ and $y(i)$ is retained at the end. To ensure the integrity of the information contained in the time series, the same process is repeated from the end of the two accumulation sequences $x(i)$ and $y(i)$, and a total of $2Ns$ segments are obtained.

Step 3: Calculate the trend-free covariance.

In each section $\lambda$, the least squares method is used to determine the fitted polynomials $x_\lambda$ and $y_\lambda$.

For $\lambda = 1, 2,...,Ns$, the detrending covariance is:

$$F^2(s, \lambda) = \frac{1}{s}\sum_{i=1}^{s}\{X[(\lambda - 1)s + i] - x_\lambda(i)\} \times \{Y[(\lambda - 1)s + i] - y_\lambda(i)\} \quad (2)$$

For $\lambda = Ns+1,...,2Ns$, the detrending covariance is:

$$F^2(s, \lambda) = \frac{1}{s}\sum_{i=1}^{s}\{X[N - (\lambda - N_S)s + i] - x_\lambda(i)\} \times \{Y[N - (\lambda - N_S)s + i] - y_\lambda(i)\} \quad (3)$$

Step 4: Obtain the q-order wave function by averaging all segments $\lambda$.

$$F_q(s) = \left\{\frac{1}{2N_s}\sum_{\lambda=1}^{2N_S}[F^2(s, \lambda)^{q/2}]\right\}^{1/q}, \quad q \neq 0 \quad (4)$$

In general, $q$ can be any nonzero number. According to the L'Hopital rule, when $q = 0$, the $q$-order wave function shown below is obtained:

$$F_0(s) = exp\left\{\frac{1}{4N_s}\sum_{\lambda=1}^{2N_S}[lnF^2(s, \lambda)]\right\}, \quad q = 0 \quad (5)$$

Step 5: Calculate the wave function $F_q(s)$ under different scales $s$. If the two time series are long-range cross-correlated, the power-law relationship can be expressed as:

$$F_q(s) \sim s^{H_{xy}(q)} \quad (6)$$

Then, take the logarithm to obtain the following formula:

$$log\,F_s(s) = H_{xy}(q)log(s) + log\,f_{()}C \quad (7)$$

where $C$ is a constant and $H_{xy}(q)$ is a generalized cross-correlation exponent that can be expressed by calculating the slope of the log plot of $F_q(s)$ versus $s$.

If $H_{xy}(q)$ varies with $q$, the cross-correlations between the two time series are multifractal. When $q = 2$, the cross-correlation exponent $H_{xy}(q)$, which is in the range $(0, 1)$, is the same as the Hurst exponent $H$. If $H = 0.5$, the two time series are not cross-correlated, which means that the change of one series in the random walk will not affect that of the other. If $0<H<0.5$, the cross-correlations between the two time series related to $q$ are antipersistent, and the increase in one series may be followed by the decline of another. If $0.5<H<1$, the cross-correlations between the two time series are persistent, and the two series have the same changing trend. In addition, in the description of $H_{xy}(q)$, $q>0$ indicates that there is a large fluctuation in the interval, and $q<0$ means the existence of a small fluctuation in the interval.

Step 6: Calculate the Renyi exponent, which can be used to characterize the multifractal properties. The relationship between $\tau(q)$ and the generalized Hurst exponent $H_{xy}(q)$ can be obtained by MF-DCCA:

$$\tau(q) = qH_{xy}(q) - 1 \quad (8)$$

If the relationship between $\tau(q)$ and $q$ is linear, then the cross-correlation between the two sequences is monofractal; otherwise, it is multifractal.

Step 7: Calculate the width $f(\alpha)$ of the multifractal spectrum derived by the Legendre transform as:

$$\alpha = H_{xy}(q) + qH'_{xy}(q) \tag{9}$$

$$f(\alpha) = q(\alpha - H_{xy}(q)) + 1 \tag{10}$$

$H'_{xy}(q)$ is the derivative of $H_{xy}(q)$ with respect to $q$. $\alpha$ is the singular intensity, describing the singularity and unity in the time series. $f(\alpha)$ is a multifractal spectrum, and its value reflects the fractal dimension of the Hölder exponent $\alpha$.

$\Delta H$ is the width of the multifractal spectrum and can be used to express the intensity of the multifractality.

$$\Delta H = H_{max}(q) - H_{min}(q) \tag{11}$$

## Data

SHIBOR (Shanghai Interbank Offered Rate), the benchmark interest rate for China's money market, is a quotation group of 18 commercial banks that are relatively active in the Chinese currency market. SHIBOR can describe the supply and demand of funds in the domestic market and has been widely used in bond issuance pricing, derivatives trading, bill transaction pricing and internal financial institution transfer pricing. This paper selects SHIBOR and Shanghai Stock Market Liquidity (SHSML) with daily data from October 9, 2006, to March 31, 2020, to detect the nonlinear relationship between the money market rate and stock market liquidity in China considering the availability of data. The daily closing price and turnover rate of the Shanghai Composite Index are used to calculate the Shanghai Stock Market Liquidity (SHSML), and they are obtained from the Wind. The data for SHIBOR (O/N) are collected from http://www.shibor.org.

Let $r_t$ be the SHIBOR at day $t$. $R_t$ is the logarithmic difference between $r_t$ and $r_{t-1}$.

$$R_t = log(r_t) - log_{f()}(r_{t-1}) \tag{12}$$

Stock market liquidity, as proposed by Amihud and Mendelson (1986) [40], is the time or cost required to complete an exchange within a certain period of time. In other words, it is the time or cost required to find an ideal price. If investors can buy or sell a large number of stocks at a lower cost without large fluctuations in the stock market, stock market liquidity is good. Motivated by the *TPI* provided by Angelidis and Andrikopoulos (2010) [41], we constructed the stock market illiquidity index as follows:

$$TPI_{id} = \frac{TO_{id}}{|R_{id}|} \tag{13}$$

Here, *TPI* is the stock market liquidity, defined as the ratio of the stock return $R_{id}$ to the turnover rate $TO_{id}$. Higher *TPI* indicates better stock market liquidity.

Fig 1 illustrates the dynamic changes in SHIBOR and SHSML, which display volatility shifts in the money market rate and stock market liquidity in China.

As shown in Table 1, excess nonzero skewness and kurtosis are exhibited in the descriptive statistics of SHIBOR and SHSML. This means that none of the series follow a normal distribution and are all fat tailed. The Jarque–Bera test formally confirms the above. At the 1% significance level, the null hypothesis of the normal distribution is rejected. This outcome indicates the complexity of the time series, confirming the difficulty in demonstrating the relationship between the money market rate and stock market liquidity in China by a linear model.

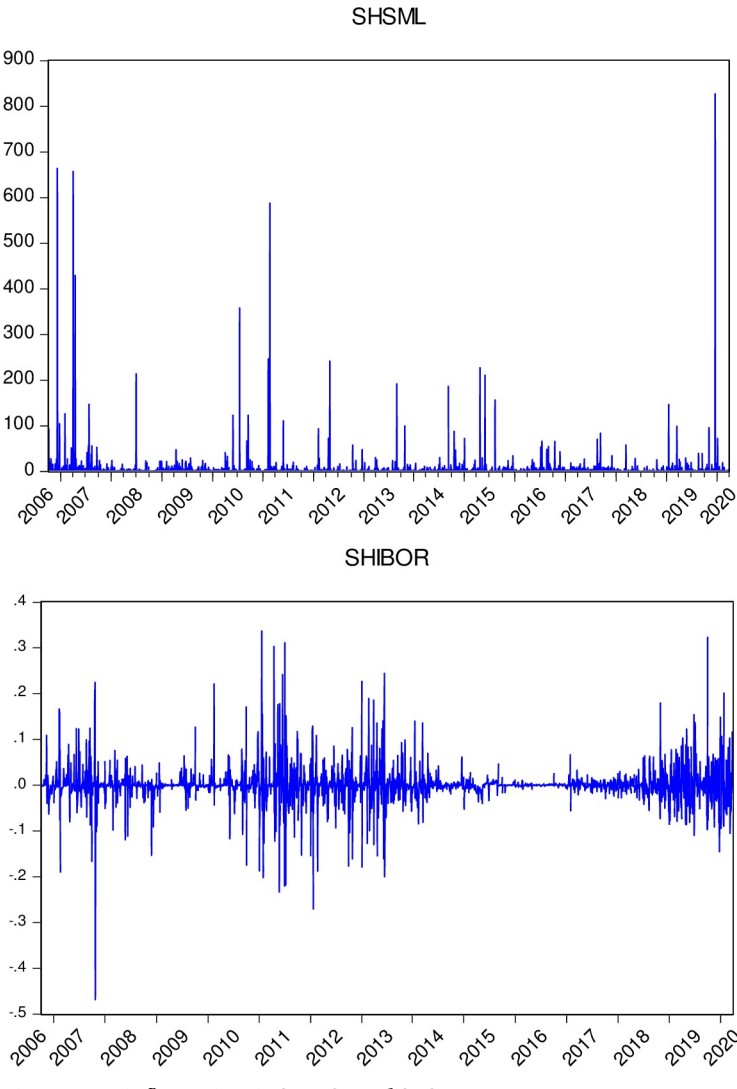

**Fig 1. Dynamic fluctuations in SHIBOR and SHSML.**

## Empirical analysis

### Cross-correlation test

This section employs the cross-correlation statistic $Q_{cc}(m)$ to qualitatively confirm the cross-correlations between the money market rate and stock market liquidity in China. Consider the

**Table 1. Descriptive statistics of SHIBOR and SHSML.**

|  | SHIBOR | SHSML |
|---|---|---|
| Mean | 0.0000840 | 4.945800 |
| Median | -0.0000871 | 1.119612 |
| Maximum | 0.336964 | 827.2934 |
| Minimum | -0.468250 | 0.035166 |
| Std. Dev. | 0.036764 | 29.31888 |
| Skewness | 0.301135 | 18.54828 |
| Kurtosis | 27.29207 | 416.2598 |
| Jarque–Bera | 80672.60 | 23521339 |

two time series $\{x_t, t = 1, 2, . . ., N\}$ and $\{y_t, t = 1, 2, . . ., N\}$. The cross-correlation statistic $Q_{cc}(m)$ is as follows:

$$Q_{CC}(m) = \sum_{i=1}^{m} \frac{c_i^2}{N - i} \tag{14}$$

The cross-correlation function is:

$$c_i = \frac{\sum_{k=i+1}^{N} x_k y_{k-i}}{\sqrt{\sum_{k=1}^{N} x_k^2 \sum_{k=1}^{N} y_k^2}} \tag{15}$$

The cross-correlation statistic $Q_{cc}(m)$ is a $\chi^2(m)$ distribution with an approximate degree of freedom of $m$. If the cross-correlation statistic $Q_{cc}(m)$ is greater than the critical value of the $\chi^2(m)$ distribution at a certain level of significance, then there are cross-correlations between the two time series; otherwise, the relationship between the two time series is not cross-correlated.

We can see from Fig 2, which shows the cross-correlation statistic $Q_{cc}(m)$ between SHIBOR and SHSML, that the value of the cross-correlation statistic $Q_{cc}(m)$ between SHIBOR and SHSML is greater than the critical value of the $\chi^2(m)$ distribution at the 5% significance level in the range of degrees of freedom from 1 to 700. This observation means that the relationship between the money market rate and stock market liquidity in China is cross-correlated.

## DCCA coefficient

To quantify the similarity between two nonstationary time series, this section adopts detrended cross-correlation analysis (DCCA) based on the detrended covariance. The DCCA coefficient $\rho_{DCCA}$, i.e., the ratio of the fluctuation function of the detrended covariance $F_{DCCA}^2$ to the product of two detrended variances $F_{DFA}$, is as follows:

$$\rho_{DCCA} = \frac{F_{DCCA}^2(s)}{F_{DCCA1}(s) F_{DCCA2}(s)} \tag{16}$$

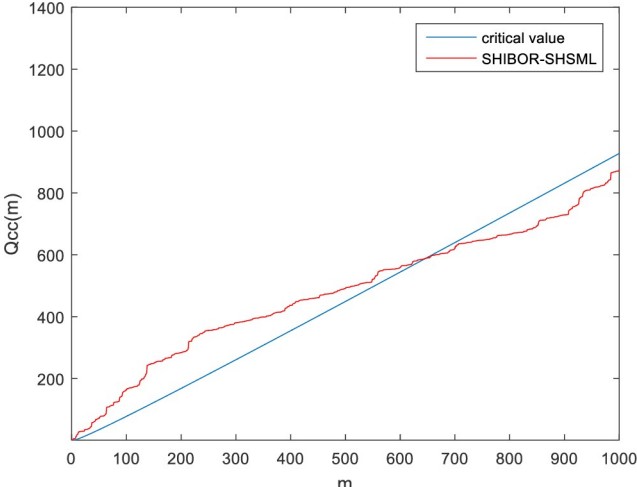

**Fig 2. Cross-correlation statistic $Q_{cc}(m)$ between SHIBOR and SHSML.**

The value of $\rho_{DCCA}$ ranges from -1 to 1. When $\rho_{DCCA} = -1$, there exist anti-cross-correlations; when $\rho_{DCCA} = 0$, there are no cross-correlations; and when $\rho_{DCCA} = 1$, there exist perfect cross-correlations. The value of the DCCA coefficient between SHIBOR and SHSML is shown in Fig 3, and the DCCA coefficients calculated for different window sizes $s$ (8, 16, 32, 64, 128, 256) are reported in Table 2. According to the range of $\rho_{DCCA}$, the value of $\rho_{DCCA}$ is not zero, and there are variations in the persistence of the cross-correlations between SHIBOR and SHSML under different window sizes $s$. These results obtained from the DCCA coefficient are consistent with the previous cross-correlation test results and further confirm the existence of power-law cross-correlations between the money market rate and stock market liquidity in China.

## MF-DCCA analysis

On the basis of the cross-correlations between the money market rate and stock market liquidity in China, this selection establishes the MF-DCCA model for SHIBOR and SHSML to quantify the cross-correlations between them and detect the characteristics of the impact of the money market rate on stock market liquidity. First, we estimate the cross-correlation exponents $H_{xy}(q)$ according to formulas (1) to (6) and obtain the logarithmic chart of the $q$-order wave function $F_q(s)$ and time scale $s$ of SHIBOR and SHSML in the range of $q = [-10, 10]$, as shown in Fig 4. For all $q$ values in the time scale $s$, the curves of the $q$-order wave function $F_q(s)$ and $logs$ are linear, suggesting that there are power-law cross-correlations between the money market rate and stock market liquidity in China. The linear trend of the curves and the cross-correlation exponent $H_{xy}(q)$ change at the time scale $s^*$ ($logs$ = 2.365 (232 days)) according to Fig 4. There is a time scale $s^*$ called the "cross-turning point", which divides the time series into the short term ($s<s^*$) and the long term ($s>s^*$).

Table 3 lists the cross-correlation exponent between SHIBOR and SHSML. When $q = 2$, the cross-correlation exponent between SHIBOR and SHSML is larger than 0.5 when $s<s^*$, indicating that the cross-correlations between the money market rate and stock market

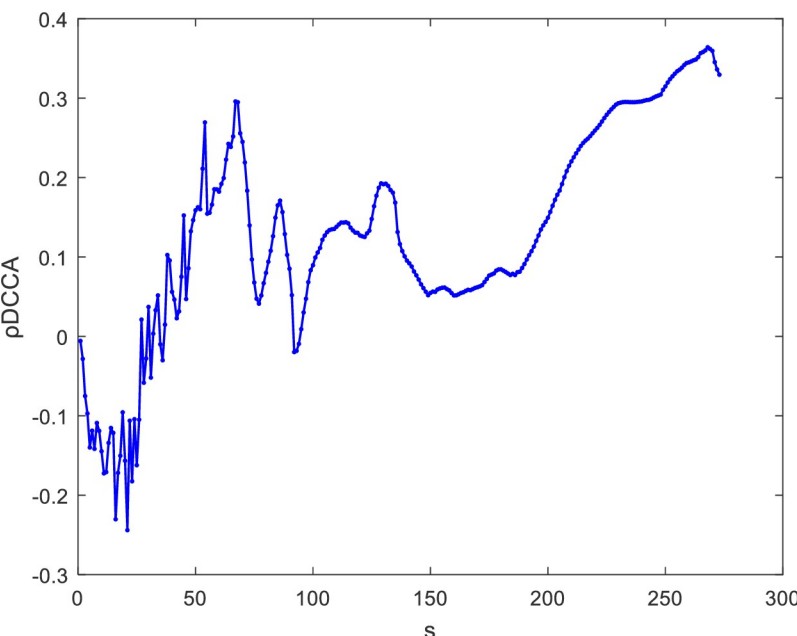

**Fig 3. The DCCA coefficient between SHIBOR and SHSML.**

**Table 2. The value of $\rho_{DCCA}$ for a given window size s.**

| Size (s) | 8 | 16 | 32 | 64 | 128 | 256 |
|---|---|---|---|---|---|---|
| $\rho_{DCCA}$ | -0.1090 | -0.2303 | 0.0034 | 0.2425 | 0.1874 | 0.3358 |

liquidity in China are persistent or have long-term memory in the short term. Moreover, the cross-correlation exponent between SHIBOR and SHSML is less than 0.5 when $s>s^*$, which means that there are antipersistent cross-correlations between the money market rate and stock market liquidity in the long term. In addition, Table 3 shows that the antipersistence between the money market rate and stock market liquidity in the long run is much greater than the persistence in the short run. Therefore, the relationship between the money market rate and stock market liquidity in China is nonlinear and dynamic. Specifically, the money market rate and stock market liquidity in China have the same changing trend in the short term, money market rate cuts will lead to an increase in stock market liquidity in the long term, and the negative impact of the money market rate on stock market liquidity is more significant than the positive implication.

At the same time, we also find that the cross-correlation exponent $H_{xy}(q)$ varies with the change in $q$ from Table 3. This result further indicates that the relationship between the money market rate and stock market liquidity in China presents multifractality. $\Delta H$, the difference between the maximum and minimum values of $H_{xy}(q)$, can be used to measure the multifractal degree of the cross-correlations between them. According to Table 3, it is not difficult to observe that the short-term multifractality strength of the cross-correlations between the money market rate and stock market liquidity is much greater than the multifractality strength in the long term, explaining the changes in the impact of the money market rate on stock market liquidity at different temporal scales. Specifically, due to unexpected exogenous emergencies and the increased complexity of the market's driving forces, the cross-correlations between the money market rate and stock market liquidity in China display stronger multifractality strength in the short run. This will affect the implementation and efficiency of the interest rate policy, causing the strange phenomenon that the negative relationship between the interest rate and stock market liquidity tends to be positive in the short run.

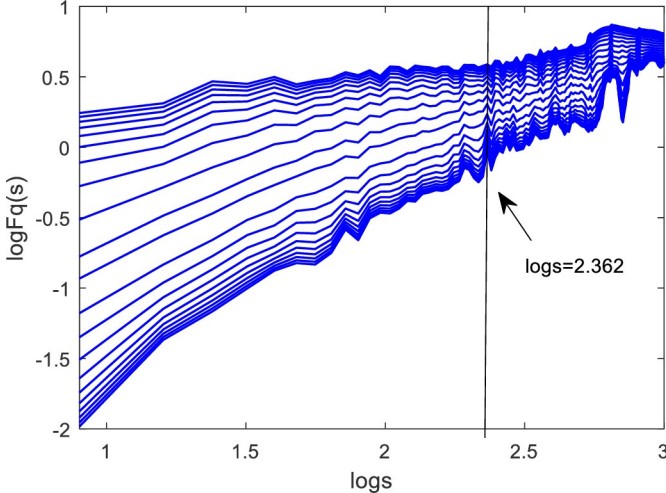

**Fig 4. Logarithmic chart of the $q$-order wave function $F_q(s)$ and time scale $s$ of SHIBOR and SHSML.**

**Table 3. The cross-correlation exponent between SHIBOR and SHSML.**

| $q$ | SHIBOR-SHSML | |
|---|---|---|
| | $s^* = 232$ | |
| | $s < s^*$ | $s > s^*$ |
| -10 | 1.140731 | 0.748766 |
| -9 | 1.130964 | 0.738285 |
| -8 | 1.11885 | 0.725573 |
| -7 | 1.103547 | 0.709874 |
| -6 | 1.083855 | 0.690086 |
| -5 | 1.058156 | 0.664632 |
| -4 | 1.024667 | 0.631368 |
| -3 | 0.982791 | 0.587888 |
| -2 | 0.93546 | 0.532977 |
| -1 | 0.884995 | 0.469435 |
| 0 | 0.777966 | 0.402925 |
| 1 | 0.718999 | 0.344485 |
| **2** | **0.587416** | **0.291792** |
| 3 | 0.468807 | 0.246744 |
| 4 | 0.384921 | 0.209186 |
| 5 | 0.327873 | 0.178515 |
| 6 | 0.287709 | 0.153733 |
| 7 | 0.258164 | 0.13372 |
| 8 | 0.235581 | 0.117457 |
| 9 | 0.217771 | 0.10411 |
| 10 | 0.203372 | 0.093029 |
| **ΔH** | **0.937359** | **0.655737** |

According to formula (8), we obtain the short-term and long-term Renyi exponents, which can further prove the existence of multifractal characteristics in the nonlinear relationship between the money market rate and stock market liquidity in China. As shown in Figs 5 and 6, the relationship between $\tau(q)$ and $q$ in the Renyi exponent is nonlinear. This result confirms that multifractality exists in the cross-correlations between the money market rate and stock market liquidity in China, leading to a better understanding of the nonlinear and dynamic relationship between the interest rate and stock market liquidity. The cross-correlations between the interest rate and stock market liquidity are complex and nonuniform, showing that there are variations in the impact of the interest rate policy on stock market liquidity at different time scales and in different spatial scenarios.

According to formulas (9) and (10), the multifractal spectrum shown in Fig 7 suggests that the multifractal spectra for series of SHIBOR and SHSML all appear as parabolic rather than as a single point, which indicates that there are multifractal characteristics in the money market rate and stock market liquidity. Moreover, the width of the multifractal spectrum can be used to express the multifractality strength. The wider the multifractal spectrum, the higher the strength of multifractality is. From Fig 7, it is not difficult to find that the multifractality strength of both the money market rate and stock market liquidity is larger than the multifractality strength of the cross-correlations between them, suggesting the macroeconomic effects of the money market rate. Although the correlations between the interest rate and stock market liquidity display multifractality, the interest rate policy is one of the most important policy tools used by major central banks in controlling stock market liquidity to create stable conditions for economic growth and sustainable development.

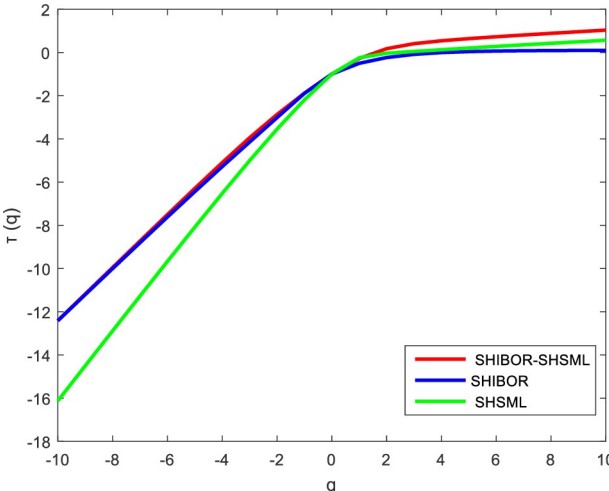

**Fig 5. The Renyi exponent between SHIBOR and SHSML in the short term.**

When $q = 2$, the cross-correlation exponent is approximately equal to the average value of each Hurst exponent in the two autoregressive fractionally integrated moving average (ARFIMA) processes, which share the same random noise. If $x(i)$ and $y(i)$ are the same time series, the cross-correlation exponent $H_{xy}(q)$ becomes the generalized Hurst index $H_{xx}(q)$ or $H_{yy}(q)$. Zhou (2008) found that the relationship between two time series constructed by the binomial measure of the p model is as follows [42]:

$$H_{xy}(q) = \frac{[H_{xx}(q) + H_{yy}(q)]}{2} \qquad (17)$$

According to formula (16), we calculate the average scaling exponent. Figs 8 and 9 report the cross-correlation exponent between SHIBOR and SHSML and their average scaling exponent. From Fig 8, when $q<0$, the cross-correlation exponent between SHIBOR and SHSML is smaller than the average scaling exponent, indicating that the positive persistence of small

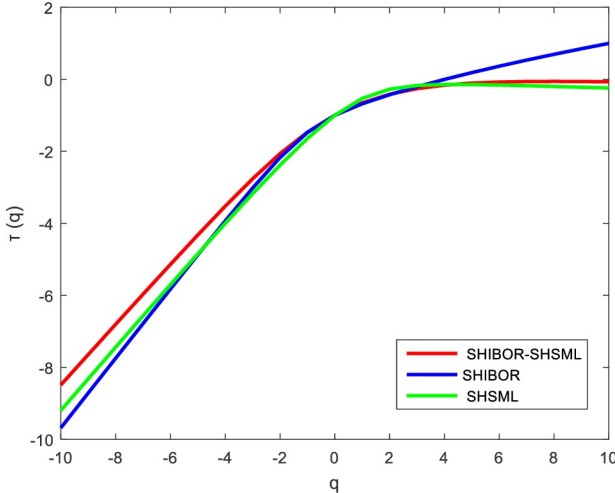

**Fig 6. The Renyi exponent between SHIBOR and SHSML in the long term.**

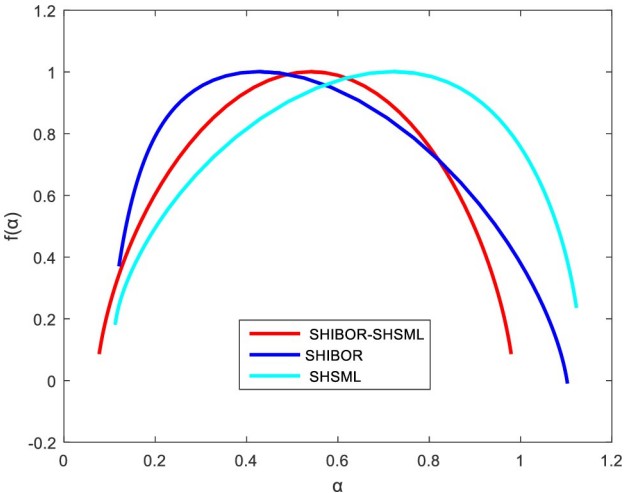

**Fig 7. The multifractal spectra for series of SHIBOR and SHSML.**

fluctuations caused by the external environment is mainly affected by the autocorrelation of the money market rate and stock market liquidity in China; as the volatility increases, the auto-correlation of the money market rate and stock market liquidity becomes weak, and the cross-correlations between them increase. When *q>0*, the cross-correlation exponent between SHI-BOR and SHSML is larger than their average scaling exponent, which means that the positive persistence of large fluctuations associated with the environment mainly arises from the cross-correlations between them in the short run. The cross-correlations between the money market rate and stock market liquidity in China are the main source of the multifractality in the two markets, and the Chinese stock market liquidity is more sensitive to fluctuations in the money market rate in the short term. From Fig 9, whether *q<0* or *q>0*, the cross-correlation exponent between SHIBOR and SHSML is less than their average scaling exponent, showing that the positive persistence of both small and large fluctuations caused by the external environment is associated with the autocorrelation of the money market rate and stock market liquidity in the long run. The Chinese stock market liquidity is inelastic in response to the money market rate

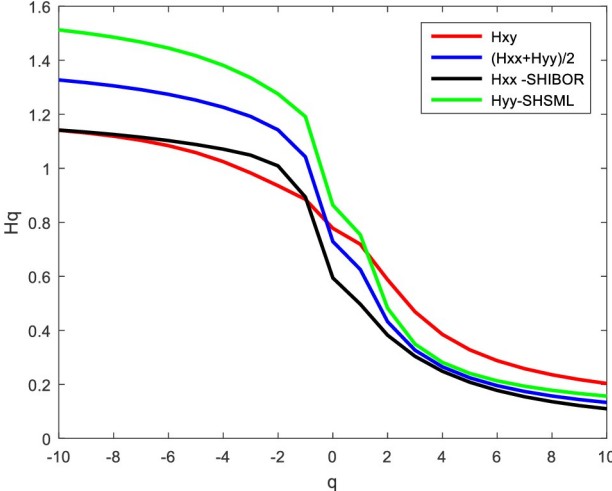

**Fig 8. Short-term scaling exponent between SHIBOR and SHSML.**

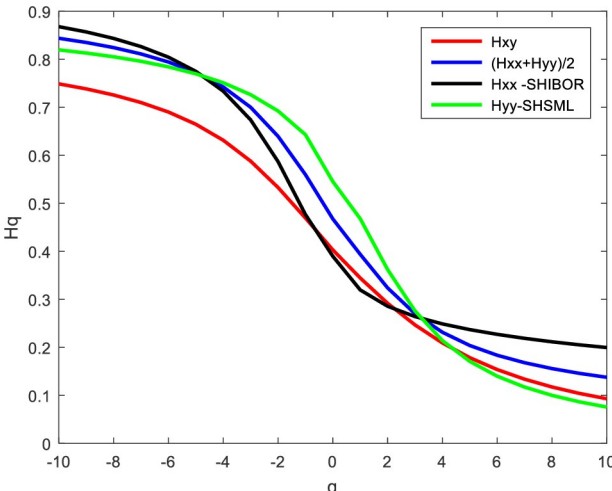

**Fig 9. Long-term scaling exponent between SHIBOR and SHSML.**

in the long term. Therefore, the money market rate has a strong ability to adjust the short-term funds of the Chinese stock market, but the guidance for long-term investment is weaker, suggesting a limited effect of the interest rate policy on stock market liquidity in the long term. These results further confirm that the interest rate policy is an emergency response rather than an effective response to mounting concerns about the lack of stock market liquidity.

## Rolling window analysis

The MF-DCCA method cannot actually capture the dynamic changes in the correlations between the money market rate and stock market liquidity in China during the entire research period because of the existence of structural interruptions or regime changes in financial time series. To identify the time-varying properties of financial variables and detect the possible impacts of exogenous events, rolling window analysis is adopted to capture the dynamic characteristics of cross-correlations between time series in this section. Since the exponent at a given time $t$ depends on the length of the time window, it is crucial for rolling window analysis to adjust the window length to an appropriate level. If the window length is too long, the locality of the calculated exponent may be hidden by the strong seasonality and periodicity of the time series, and the effects of external events on short-term market dynamics will be obscured. If the window length is too short, it will be difficult to observe the trend of the exponent. According to Wang (2011) [43] and Ruan (2016) [44], we select 250 days as the window length to avoid localized loss and overvolatility of the cross-correlation exponents. The cross-correlation exponents calculated by rolling window analysis are reported in Fig 10.

Fig 10 shows that the scaling exponent between SHIBOR and SHSML is almost greater than 0.5, indicating a positive impact of the money market rate on stock market liquidity in the short term. As the window moves, the scaling exponent between SHIBOR and SHSML is time-varying, and stronger volatility in the scaling exponent between SHIBOR and SHSML has been shown during the financial crises in 2008 and the stock market crash in 2015. When crises and emergencies occur, there are stronger positive cross- correlations between interest rates and stock market liquidity, and investors are more sensitive to changes in the interest rate. However, the guidance of the interest rate for investors in the long run is weak; if policy-makers continually cut the interest rate in unexpected exogenous emergencies, the opposite effect will occur. For example, the efficiency of interest rate cuts in reversing the turbulent

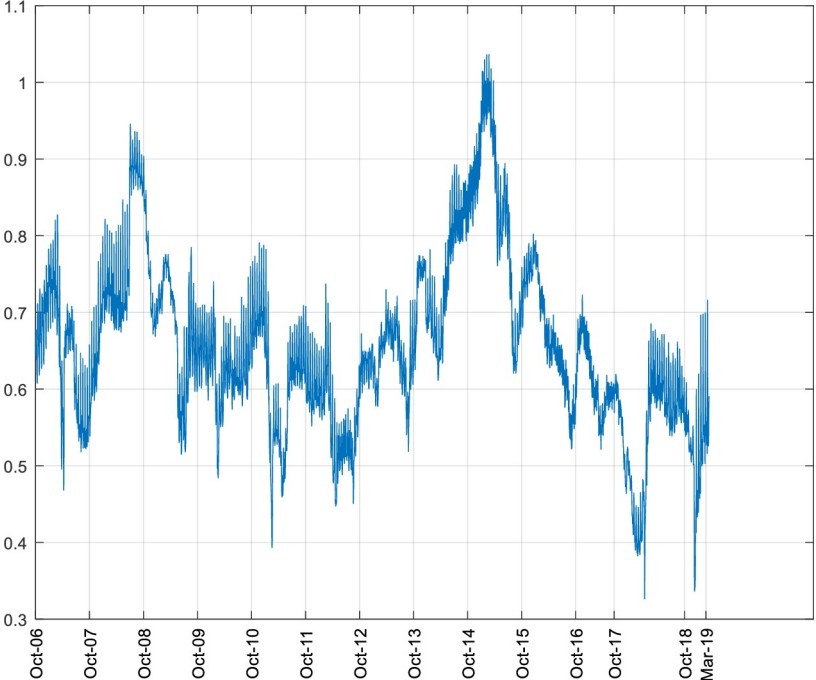

**Fig 10. Dynamic changes in the scaling exponents when q = 2.**

downward stock market liquidity caused by the public health emergency in 2020 is still unknown. As Wang (2020) [45] documented, a low interest rate can be a source of global system risk, especially in the short run.

## Conclusions

This paper employs the MF-DCCA model to demonstrate the nonlinear relationship between the money market rate and stock market liquidity in China from a multifractal perspective, explaining the inconsistent findings obtained by previous studies regarding the variations in the real relationship between the interest rate and stock market liquidity. The main relationship between the money market rate and stock market liquidity in China is as follows.

There are cross-correlations between the money market rate and stock market liquidity in China. The cross-correlations show positive persistence in the short run and antipersistence in the long run, and the negative cross-correlations between the interest rate and stock market liquidity are more significant than the positive cross-correlations between them. This result suggests variations in direction in the cross-correlations between the interest rate and stock market liquidity. As Jensen and Moorman (2010) [46] and Fernández-Amador et al. (2013) [47] noted, the relationship between the interest rate and stock market liquidity is nonlinear and dynamic. Further, the cross-correlations between the money market rate and stock market liquidity in China also display multifractality, leading to a better understanding of the nonlinear relationship between the interest rate and stock market liquidity. The multifractality strength of the cross-correlations between the money market rate and stock market liquidity in the short term is much greater than the multifractality strength in the long term, providing an explanation for the changes in direction of the impact of the interest rate on stock market liquidity at different temporal scales.

The multifractal spectrum shows that the multifractality strength of both the money market rate and stock market liquidity in China is larger than the multifractality strength of the cross-correlations between them, suggesting the macroeconomic effects of the money market rate. Although multifractality exists in the cross-correlations between the interest rate and stock market liquidity, it is still effective for interest rate policy to control stock market liquidity. Moreover, according to the changes in the cross-correlation exponent between the money market rate and stock market liquidity and their average scaling exponent, we find that the cross-correlations between the money market rate and stock market liquidity in the short run are the main source of the multifractality in the two markets and that their autocorrelation causes the multifractality of the two markets in the long run. The results indicate the limitation of interest rate adjustment efficiency in the long term. The interest rate policy is an emergency response rather than an effective response to mounting concerns regarding the lack of stock market liquidity.

Rolling window analysis indicates that the scaling exponent between the money market rate and stock market liquidity in China shows stronger volatility during the financial crises in 2008 and the stock market crash in 2015. The positive cross-correlations between the money market rate and stock market liquidity in the short run become strong in periods of crises and emergencies. Interest rate policy is an emergency technique used by major central banks to add additional stock market liquidity during periods of increased economic uncertainty; however, the interest rate policy in bailout plans after a crisis will not be as effective as expected if the interest rate is cut continually.

## Acknowledgments

The authors thank all the participants for giving invaluable responses in this study.

## Author Contributions

**Writing – original draft:** Yihong Sun.

**Writing – review & editing:** Xuemei Yuan.

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
