## [Decision Letter · Decision Letter 0]

2 Dec 2020

PONE-D-20-32997

Nonlinear Relationship between SHIBOR and Chinese Stock Market Liquidity: A Multifractal Analysis

PLOS ONE

Dear Dr. Yuan,

Thank you for submitting your manuscript to PLOS ONE. After careful consideration, I feel that it has merit but does not fully meet PLOS ONE’s publication criteria as it currently stands. Therefore, I invite you to submit a revised version of the manuscript that addresses the points raised during the review process.

Reviewers consider that the paper is suitable of publication after some minor corrections. Major concerns are showed by reviewer 2, so I must suggest to the authors to focus on his comments. I agree with him that some points need more clarification. Reviewers propose to include some relevant literature but I will suggest also to read a recent publication about some problems of the multifractal processes. See the DOI 10.1016/j.chaos.2020.109914

We look forward to receiving your revised manuscript.

Kind regards,

J E. Trinidad Segovia

Academic Editor

PLOS ONE

Journal Requirements:

Reviewers' comments:

Reviewer's Responses to Questions

**Comments to the Author**

1. Is the manuscript technically sound, and do the data support the conclusions?

Reviewer #1: Yes

Reviewer #2: Yes

Reviewer #3: Yes

2. Has the statistical analysis been performed appropriately and rigorously? 

Reviewer #1: Yes

Reviewer #2: Yes

Reviewer #3: Yes

3. Have the authors made all data underlying the findings in their manuscript fully available?

Reviewer #1: No

Reviewer #2: Yes

Reviewer #3: No

4. Is the manuscript presented in an intelligible fashion and written in standard English?

Reviewer #1: Yes

Reviewer #2: No

Reviewer #3: Yes

5. Review Comments to the Author

Reviewer #1: For me, this paper is quite clear-cut. The MF-DCCA method is correctly employed. The only comment is that the authors need to cite some relevant on MF-DCCA, especially in the Chinese stock market. For example:

Xiong, X., Xu, K., & Shen, D. (2019). Dynamic Cross-Correlations between Investors’ Attention and CSI300 Index Futures. Fluctuation and Noise Letters, 18(04), 1950022. doi: 10.1142/S0219477519500226

Zhang, W., Wang, P., Li, X., & Shen, D. (2018). The inefficiency of cryptocurrency and its cross-correlation with Dow Jones Industrial Average. Physica A: Statistical Mechanics and its Applications, 510, 658-670. doi: https://doi.org/10.1016/j.physa.2018.07.032

Zhang, W., Wang, P., Li, X., & Shen, D. (2018). Multifractal Detrended Cross-Correlation Analysis of the Return-Volume Relationship of Bitcoin Market. Complexity, 8691420, 1-20. doi: 10.1155/2018/8691420

Zhang, Z., Zhang, Y., Shen, D., & Zhang, W. (2018). The Dynamic Cross-Correlations between Mass Media News, New Media News, and Stock Returns. Complexity(7619494), 11. doi: 10.1155/2018/7619494

Reviewer #2: The paper represents some interesting data but the representation is not up to the mark and there are various ambiguities.

1. The full form of SHIBOR should be clarified in the abstract/introduction. One should be clear about the variables that are discussed in the paper. The author mentions:

“This paper employs the Multifractal Detrended Cross-Correlation Analysis (MF9 DCCA) model to estimate the nonlinear relationship between SHIBOR and Chinese 10 stock market liquidity….” throughout the abstract and introduction and finally in page 10 one comes to know about the variables whose cross correlations have been studied! It should be explained without ambiguity from the very beginning for the sake of interest of the readers. One needs to Google to know what exactly SHIBOR means. Not everyone is based in China.

2.The authors have chosen a period from Oct 2006 to March 2020. Kindly mention why this period was chosen? Is the data before 2006 not available? Generally for a statistical analysis length of data is very important.

3.in Fig 3 mention which plot is for which value of q

4.It is evident from Fig. 6 that the multifractal spectrum for SHIBOR is skewed to the left while for SHSML it is skewed to the right. Explain this finding.

5. The details of the rolling window analysis should be provided. The authors have simply mentioned the length of the window. Was analysis carried out for each and every data point? What was the range of s for the rolling window analysis? For the complete data set cross over occurs at 232 days. Is there any connection between choosing 250 days as the window size?

6.Fig 9: The values in the plot are not readable

7. The paper contains many typographical errors which should be corrected.

Reviewer #3: Review report on “Nonlinear Relationship between SHIBOR and Chinese Stock Market Liquidity: A Multifractal Analysis”:

1. The literature on the new progress and application of the DFA, DCCA, MF-DFA and MF-DCCA family should be updated. I listed some examples as follows:

https://journals.aps.org/pre/abstract/10.1103/PhysRevE.84.016106

https://journals.aps.org/pre/abstract/10.1103/PhysRevE.84.066118

https://journals.aps.org/pre/abstract/10.1103/PhysRevE.91.062816

https://www.sciencedirect.com/science/article/abs/pii/S0378437110008800

https://www.sciencedirect.com/science/article/abs/pii/S0378437117307781

https://www.sciencedirect.com/science/article/abs/pii/S0378437113003403

http://www.nature.com/doifinder/10.1038/srep08143

2. The authors used the MF-DCCA approach to study the nonlinear correlations between SHIBOR and Chinese stock market liquidity. This approach has been widely used to study a pair of financial entities. Thus, the new contributions of this manuscript are limited. I would like to suggest the authors use other new approach to study the nonlinear relationship, such as the DCCA coefficient and detrended partial cross-correlation analysis.

3. The authors should point out the source of the data.

4. The authors should state the motivations of choosing October 9, 2006 and March 31, 2020 as the beginning date and the ending date of the sample period.

5. There are several grammatical errors and typos. The manuscript could profit from more thorough editing and proof-reading.

6. PLOS authors have the option to publish the peer review history of their article (what does this mean?). If published, this will include your full peer review and any attached files.

Reviewer #1: No

Reviewer #2: No

Reviewer #3: No

---

## [Author Response · Author response to Decision Letter 0]

5 Jan 2021

PONE-D-20-32997

EMID: ce6b938cca08ff01

Original Title: Nonlinear Relationship between SHIBOR and Chinese Stock Market Liquidity: A Multifractal Analysis

Title: Nonlinear Relationship between Money Market Rate and Stock Market Liquidity in China: A Multifractal Analysis

PLOS ONE

Dear Editor and the anonymous reviewers:

Please find enclosed our manuscript, which has been thoroughly revised following your comments and suggestions. We believe that each and every concern raised in your reviews of the paper’s original version has been satisfactorily resolved. We also believe that the paper is now in much better shape as a consequence and appreciate very much all the comments made on the paper. We highlighted the changes to our manuscript using colored text. You will also find a complete list of the detailed changes we have made in response to the reviewers’ comments below, which are presented on a point-by-point basis. The following is a list of the main changes made in this revision.

1. In response to reviewer 1 and reviewer 3, we introduce literature on the applications of the MF-DCCA approach to stock markets and the new progress and application of the DFA, DCCA, MF-DFA and MF-DCCA methods.

2. As suggested by reviewer 2 and reviewer 3, the new manuscript has been edited and proofread.

3. Response to reviewer 2’s concerns, we replaced “SHIBOR” with “money market rate” in the title and new manuscript to clarify the variables that are discussed in the paper. 

4. Based on the suggestions of reviewer 2, we offer clearer details of the rolling window analysis, which result in a better understanding of the process.

5. According to reviewer 3’s advice, this revision adds the DCCA coefficient section to comprehensively examine the nonlinear correlations between money market rate and stock market liquidity through another new approach.

Overall, we have responded to all arguments, recommendations, and suggestions from the referees in the new version. We hope that you agree with our responses. Again, we appreciate all of your comments and kind considerations regarding this paper. Thank you very much.

Sincerely yours,

Xuemei Yuan

 

Reviewer #1: For me, this paper is quite clear-cut. The MF-DCCA method is correctly employed. The only comment is that the authors need to cite some relevant on MF-DCCA, especially in the Chinese stock market. For example:

Xiong, X., Xu, K., & Shen, D. (2019). Dynamic Cross-Correlations between Investors’ Attention and CSI300 Index Futures. Fluctuation and Noise Letters, 18(04), 1950022. doi: 10.1142/S0219477519500226

Zhang, W., Wang, P., Li, X., & Shen, D. (2018). The inefficiency of cryptocurrency and its cross-correlation with Dow Jones Industrial Average. Physica A: Statistical Mechanics and its Applications, 510, 658-670. doi: https://doi.org/10.1016/j.physa.2018.07.032

Zhang, W., Wang, P., Li, X., & Shen, D. (2018). Multifractal Detrended Cross-Correlation Analysis of the Return-Volume Relationship of Bitcoin Market. Complexity, 8691420, 1-20. doi: 10.1155/2018/8691420

Zhang, Z., Zhang, Y., Shen, D., & Zhang, W. (2018). The Dynamic Cross-Correlations between Mass Media News, New Media News, and Stock Returns. Complexity(7619494), 11. doi: 10.1155/2018/7619494

Authors’ reply:

Thank you for your suggestion. Clearly, this manuscript did not sufficiently sum up the research on the application of the MF-DCCA method. We have cited the relevant studies on the application of MF-DCCA in the Chinese stock market in this new manuscript. “More recently, the MF-DCCA method has become widely used for earthquake and hydrology studies, especially in relation to financial markets [34,35]. Zhang et al. (2018) [36] and Xiong et al. (2019) [37] introduced the MF-DCCA method to study the Chinese stock market. Zhang et al. (2018) [38] and Zhang et al. (2018) [39] adopted the MF-DCCA method to test the efficiency of the cryptocurrency market and investigated the cross-correlations of the return-volume relationship of the cryptocurrency market.”Please see details on pages 6-7.

Zhang Z., Zhang Y., Shen D. The dynamic cross-correlations between mass media news, new media news, and stock returns. Complexity. 2018;11:7619494. doi: https://doi.org/10.1155/2018/7619494

Xiong X., Xu K., Shen D. Dynamic Cross-Correlations between Investors’ Attention and CSI300 Index Futures. Fluctuation and Noise Letters. 2019;18(04):1950022. doi: https://doi.org/10.1142/S0219477519500226

Zhang W., Wang P., Li X., Shen D. The inefficiency of cryptocurrency and its cross-correlation with Dow Jones Industrial Average. Physica A: Statistical Mechanics and its Applications. 2018;510:658-670. doi: https://doi.org/10.1016/j.physa.2018.07.032

Zhang W., Wang P., Li X., Shen D. Multifractal detrended cross-correlation analysis of the return-volume relationship of bitcoin market. Complexity. 2018:8691420. doi: https://doi.org/10.1155/2018/8691420 Please see details on page 34.

 

Reviewer #2: The paper represents some interesting data but the representation is not up to the mark and there are various ambiguities.

1. The full form of SHIBOR should be clarified in the abstract/introduction. One should be clear about the variables that are discussed in the paper. The author mentions:

“This paper employs the Multifractal Detrended Cross-Correlation Analysis (MF9 DCCA) model to estimate the nonlinear relationship between SHIBOR and Chinese 10 stock market liquidity….” throughout the abstract and introduction and finally in page 10 one comes to know about the variables whose cross correlations have been studied! It should be explained without ambiguity from the very beginning for the sake of interest of the readers. One needs to Google to know what exactly SHIBOR means. Not everyone is based in China.

Authors’ reply:

Thank you for pointing out this problem. We replaced “SHIBOR” with “money market rate” in title and the new manuscript to clarify the variable that is discussed in the paper. In specifically, we changed the title into “Nonlinear Relationship between Money Market Rate and Stock Market Liquidity in China: A Multifractal Analysis” to led to a better understanding of the topic we discussed. Please see details on page 1 In addition, we also provide an instruction about the variable in the Data section. “SHIBOR (Shanghai Interbank Offered Rate), the benchmark interest rate for China's money market, is a quotation group of 18 commercial banks that are relatively active in the Chinese currency market. SHIBOR can describe the supply and demand of funds in the domestic market and has been widely used in bond issuance pricing, derivatives trading, bill transaction pricing and internal financial institution transfer pricing. SHIBOR is a variable that represents the money market rate.”Please see details on page 11.

2. The authors have chosen a period from Oct 2006 to March 2020. Kindly mention why this period was chosen? Is the data before 2006 not available? Generally for a statistical analysis length of data is very important.

Authors’ reply:

Thank you for mentioning these concerns. The data for SHIBOR (Shanghai Interbank Offered Rate), collected from http://www.shibor.org, date back to October 2006. Therefore, we selected a period in which the SHIBOR and the Shanghai Stock Market Liquidity (SHSML) data exist at the same time to examine the characteristics of the nonlinear relationship between money market rate and stock market liquidity in China. The period includes all information related to SHIBOR and stock market liquidity. We add a footnote to explain the period we selected in Data section. Please see details on page 11.

3. In Fig 3 mention which plot is for which value of q

Authors’ reply:

Thank you for your careful review. The longitudinal axis of this figure displays the logarithmic of the q-order wave function Fq(s) in the range of q = [-10, 10]. The q-order wave function Fq(s) is calculated according to formula (4). The changes in the curves between the q-order wave function Fq(s) and logs can help us to divide the time series into long-term and short-term periods. We offer clearer details of the q-order wave function Fq(s). Please see details on pages 16-17.

4. It is evident from Fig. 6 that the multifractal spectrum for SHIBOR is skewed to the left while for SHSML it is skewed to the right. Explain this finding.

Authors’ reply:

Thank you for pointing out this problem. The multifractal spectrum indicates the probability distribution of fractal dimensions. The multifractal spectrum is usually used to express the multifractality strength of time series by comparing the width of the multifractal spectrum. Therefore, this manuscript employs the multifractal spectrum to compare the multifractality strength of each individual series and the cross-correlations between them. As for the skewness of individual series in multifractal spectrum, the multifractal spectrum for SHIBOR is skewed to the left, implying an increase in SHIBOR returns in the future. SHSML is skewed to the right, suggesting that there exists a decreasing trend in stock market liquidity in the future.

5. The details of the rolling window analysis should be provided. The authors have simply mentioned the length of the window. Was analysis carried out for each and every data point? What was the range of s for the rolling window analysis? For the complete data set cross over occurs at 232 days. Is there any connection between choosing 250 days as the window size?

Authors’ reply:

I apologize for the lack of clarity in the explanation about the rolling window analysis. We reply to each your concerns separately in the following.

(1) The details of the rolling window analysis should be provided. The authors have simply mentioned the length of the window.

Authors’ reply:

Thank you for your suggestions. This new manuscript has provided an introduction to rolling window analysis, helping to lead to a better understanding of this section. The cross-correlations between money market rate and stock market liquidity are dynamic. However, the MF-DCCA method cannot actually capture the dynamic changes in the correlations between money market rate and stock market liquidity during the entire research period because of the existence of structural interruptions or regime changes in financial time series. To identify the time-varying properties of financial variables and detect the possible impacts of exogenous events, rolling window analysis is adopted to capture the dynamic characteristics of the cross-correlations between time series in this section. Please see details on pages 24-25.

(2) Was analysis carried out for each and every data point?

Authors’ reply:

Rolling window analysis is adopted to identify the time-varying properties of financial variables and detect the possible impacts of exogenous events. It is not necessary to perform an analysis for each and every data point. As shown in the figure related to rolling window analysis, stronger volatility in the scaling exponent between money market rate and stock market liquidity during the financial crises in 2008 and the stock market crash in 2015 is notable. According to this strange phenomenon, we find that when crises and emergencies occur, there are stronger positive cross-correlations between the interest rate and stock market liquidity, and investors are more sensitive to changes in the interest rate. If a policymaker continually cuts the interest rate in unexpected exogenous emergencies, the opposite effect will occur. Please see details on pages 24-25.

(3) What was the range of s for the rolling window analysis?

Authors’ reply:

We select 250 days as the window length. Since the exponent at a given time t depends on the length of the time window, it is crucial to adjust the window length to an appropriate level for rolling window analysis. If the window length is too long, the locality of the calculated exponent may be hidden by the strong seasonality and periodicity of the time series, and the effects of external events on short-term market dynamics will be obscured. If the window length is too short, it will be difficult to observe the trend of the exponent. When the window length is 250 days, the localized loss and overvolatility of the cross-correlation exponents are avoided according to Wang (2011) and Ruan (2016) .

Wang Y., Wei Y., Wu C. Analysis of the efficiency and multifractality of gold markets based on multifractal detrended fluctuation analysis. Physica A. 2011;390(5):817-827. doi: 10.1016/j.physa.2010.11.002

Ruan Q.S., Wang, Y., Lu. X.S., Qin, J. Cross-correlations between Baltic Dry index and crude oil prices. Physica A. 2016;453(1):278-289. doi: https://doi.org/10.1016/j.physa.2016.02.018

 (4) For the complete data set cross over occurs at 232 days. Is there any connection between choosing 250 days as the window size?

Authors’ reply:

There is no connection between setting the crossover at 232 days and choosing 250 days as the window size. In MF-DCCA analysis, we employ the MF-DCCA model to study the nonlinear relationship between money market rate and stock market liquidity in the long run and short run. According to Fig 4 (the logarithmic chart of the function Fq(s) and time scale s of SHIBOR and SHSML, there is a time scale s* (logs=2.365 (232 days)) that divides the time series into the short term (s<s*) and the long term (s>s*). However, we use rolling window analysis to identify the time-varying properties of financial variables and detect the possible impacts of exogenous events. Since the exponent at a given time t depends on the length of the time window, we select 250 days as the window length to avoid localized loss and overvolatility of the cross-correlation exponents according to Wang (2011) and Ruan (2016).

Wang Y., Wei Y., Wu C. Analysis of the efficiency and multifractality of gold markets based on multifractal detrended fluctuation analysis. Physica A. 2011;390(5):817-827. doi: 10.1016/j.physa.2010.11.002

Ruan Q.S., Wang, Y., Lu. X.S., Qin, J. Cross-correlations between Baltic Dry index and crude oil prices. Physica A. 2016;453(1):278-289. doi: https://doi.org/10.1016/j.physa.2016.02.018

6. Fig 9: The values in the plot are not readable

Authors’ reply:

Thank you for your careful review. In the process of adjusting the article, this figure was modified. This part of Fig 10 is now readable. Please see details on page 25.

Fig 10 Dynamic changes in the scaling exponents when q=2.

7. The paper contains many typographical errors which should be corrected.

Authors’ reply:

I apologize for the typographical errors in the manuscript. The new manuscript has been edited and proofread according to your advice.

 

Reviewer #3:

1. The literature on the new progress and application of the DFA, DCCA, MF-DFA and MF-DCCA family should be updated. I listed some examples as follows:

https://journals.aps.org/pre/abstract/10.1103/PhysRevE.84.016106

https://journals.aps.org/pre/abstract/10.1103/PhysRevE.84.066118

https://journals.aps.org/pre/abstract/10.1103/PhysRevE.91.062816

https://www.sciencedirect.com/science/article/abs/pii/S0378437110008800

https://www.sciencedirect.com/science/article/abs/pii/S0378437117307781

https://www.sciencedirect.com/science/article/abs/pii/S0378437113003403

http://www.nature.com/doifinder/10.1038/srep08143

Authors’ reply:

Thank you for your valuable suggestions. We have introduced the literature on the new progress and application of the DFA, DCCA, MF-DFA and MF-DCCA methods in the new manuscript. For example, there are a number of techniques and approaches related to MF-DCCA to detect the long-term dependence and quantify cross-correlations between nonstationary time series. “In addition, there are a number of techniques and approaches related to MF-DCCA to detect the long-term dependence and quantify cross-correlations between nonstationary time series. Zebende (2011) [25] proposed a new coefficient that is defined in terms of the DFA method and the DCCA method with the objective of quantifying the level of cross-correlation between nonstationary time series. Podobnik (2011) [26] used DCCA and proposed the detrended cross-correlation coefficient (ρDCCA(T, n)) to quantify the cross-covariance and the cross-correlation of two nonstationary time series. Jiang and Zhou (2011) [27] developed a class of MF-DCCA algorithms based on detrending moving-average analysis, called MF-XDMA. Wang et al. (2013) [28] used the empirical cross-correlation matrices constructed by the DCCA coefficient to show properties at different time scales in the US stock market. Yuan et al. (2015) [29] proposed detrended partial cross-correlation analysis (DPCCA) based on DCCA to quantify the relations of two nonstationary signals (with the influences of other signals removed) on different time scales. Then, Lin et al. (2018) [30] employed the detrended cross-correlation analysis (DCCA) coefficient and the detrended partial cross-correlation analysis (DPCCA) coefficient to investigate cross-correlations and net cross-correlations among five major world gold markets. Qian et al. (2015) [31] analyzed multifractal binomial measures masked with strong white noise and found that the MF-DPXA method quantifies the hidden multifractal nature.”Please see details on pages 6-7.

Zebende G.F. DCCA cross-correlation coefficient: Quantifying level of cross-correlation. Physica A: Statistical Mechanics and its Applications. 2011;390(4):614-618. doi: https://doi.org/10.1016/j.physa.2010.10.022

Podobnik B., Jiang Z.Q., Zhou W.X., Stanley H.E. Statistical tests for power-law cross-correlated processes. Physical Review E. 2011; 84:066118. doi: 10.1103/PhysRevE.84.066118

Jiang Z.Q., Zhou W.X. Multifractal detrending moving-average cross-correlation analysis. Physical Review E. 2011;84 (1 Pt 2):066118. doi: 10.1103/PhysRevE.84.016106

Wang G.J., Xie C., Chen S., Yang J.J., Yang M.Y. Random matrix theory analysis of cross-correlations in the US stock market: Evidence from Pearson’s correlation coefficient and detrended cross-correlation coefficient. Physica A: Statistical Mechanics and its Applications. 2013;392(17):3715-3730. doi: https://doi.org/10.1016/j.physa.2013.04.027

Yuan N., Fu Z., Zhang H., Piao L., Xoplaki E., Luterbacher J. Detrended partial-cross-correlation analysis: A new method for analyzing correlations in complex system. Scientific Reports. 2015;5:8143. doi: 10.1103/PhysRevLett.100.084102

Lin M., Wang G.J., Xie C., Stanley H.E. Cross-correlations and influence in world gold markets. Physica A: Statistical Mechanics and its Applications.490(15):504-512. doi: https://doi.org/10.1016/j.physa.2017.08.045

Qian X.Y., Liu Y.M., Jiang Z.Q., Podobnik B., Zhou W.X., Stanley H.E. Detrended partial cross-correlation analysis of two nonstationary time series influenced by common external forces. Physical Review E. 2015;91(6):062816. doi: 10.1103/PhysRevE.91.062816 Please see details on pages 32-33.

2. The authors used the MF-DCCA approach to study the nonlinear correlations between SHIBOR and Chinese stock market liquidity. This approach has been widely used to study a pair of financial entities. Thus, the new contributions of this manuscript are limited. I would like to suggest the authors use other new approach to study the nonlinear relationship, such as the DCCA coefficient and detrended partial cross-correlation analysis.

Authors’ reply:

Thank you for pointing this out. Detrended cross-correlation analysis (DCCA) is a good approach to study the nonlinear relationship between money market rate and the stock market liquidity in China. According to your advice, we have made some adjustments and added the DCCA coefficient section to further confirm the cross-correlations between money market rate and stock market liquidity. The DCCA coefficient section adopts detrended cross-correlation analysis (DCCA) based on detrended covariance to quantify the similarity of two nonstationary time series. According to the range of ρDCCA, the value of ρDCCA is not zero. There are variations in the persistence of the cross-correlations between money market rate and stock market liquidity under different window sizes s. These results obtained from the DCCA coefficient are consistent with the previous cross-correlation test results and further confirm the existence of power-law cross-correlations between money market rate and stock market liquidity. Please see details on pages 15-16.

3. The authors should point out the source of the data.

Authors’ reply:

Thank you for your careful review. Clearly, this information was not presented clearly enough. The daily closing price and turnover rate of the Shanghai Composite Index are used to calculate the Shanghai Stock Market Liquidity (SHSML), and they are obtained from Wind. The data for SHIBOR (O/N) are collected from http://www.shibor.org. We have pointed out the sources of the data we used in the Data section. Please see details on page 11.

4. The authors should state the motivations of choosing October 9, 2006 and March 31, 2020 as the beginning date and the ending date of the sample period.

Authors’ reply:

Thank you for mentioning these concerns. The data for SHIBOR (Shanghai Interbank Offered Rate), collected from http://www.shibor.org, date back to October 2006. Therefore, we selected a period in which the SHIBOR and the Shanghai Stock Market Liquidity (SHSML) data exist at the same time to examine the characteristics of the nonlinear relationship between money market rate and stock market liquidity in China. The period includes all information related to SHIBOR and stock market liquidity. We add a footnote to explain the period we selected in Data section. Please see details on page 11.

5. There are several grammatical errors and typos. The manuscript could profit from more thorough editing and proof-reading.

Authors’ reply:

I apologize for the typographical errors in the manuscript. The new manuscript has been edited and proofread according to your advice.

Reference

1. Zhang Z., Zhang Y., Shen D. The dynamic cross-correlations between mass media news, new media news, and stock returns. Complexity. 2018;11:7619494. doi: https://doi.org/10.1155/2018/7619494

2. Xiong X., Xu K., Shen D. Dynamic Cross-Correlations between Investors’ Attention and CSI300 Index Futures. Fluctuation and Noise Letters. 2019;18(04):1950022. doi: https://doi.org/10.1142/S0219477519500226

3. Zhang W., Wang P., Li X., Shen D. The inefficiency of cryptocurrency and its cross-correlation with Dow Jones Industrial Average. Physica A: Statistical Mechanics and its Applications. 2018;510:658-670. doi: https://doi.org/10.1016/j.physa.2018.07.032

4. Zhang W., Wang P., Li X., Shen D. Multifractal detrended cross-correlation analysis of the return-volume relationship of bitcoin market. Complexity. 2018:8691420. doi: https://doi.org/10.1155/2018/8691420

5. Wang Y., Wei Y., Wu C. Analysis of the efficiency and multifractality of gold markets based on multifractal detrended fluctuation analysis. Physica A. 2011;390(5):817-827. doi: 10.1016/j.physa.2010.11.002

6. Ruan Q.S., Wang, Y., Lu. X.S., Qin, J. Cross-correlations between Baltic Dry index and crude oil prices. Physica A. 2016;453(1):278-289. doi: https://doi.org/10.1016/j.physa.2016.02.018

7. Zebende G.F. DCCA cross-correlation coefficient: Quantifying level of cross-correlation. Physica A: Statistical Mechanics and its Applications. 2011;390(4):614-618. doi: https://doi.org/10.1016/j.physa.2010.10.022

8. Podobnik B., Jiang Z.Q., Zhou W.X., Stanley H.E. Statistical tests for power-law cross-correlated processes. Physical Review E. 2011; 84:066118. doi: 10.1103/PhysRevE.84.066118

9. Jiang Z.Q., Zhou W.X. Multifractal detrending moving-average cross-correlation analysis. Physical Review E. 2011;84 (1 Pt 2):066118. doi: 10.1103/PhysRevE.84.016106

10. Wang G.J., Xie C., Chen S., Yang J.J., Yang M.Y. Random matrix theory analysis of cross-correlations in the US stock market: Evidence from Pearson’s correlation coefficient and detrended cross-correlation coefficient. Physica A: Statistical Mechanics and its Applications. 2013;392(17):3715-3730. doi: https://doi.org/10.1016/j.physa.2013.04.027

11. Yuan N., Fu Z., Zhang H., Piao L., Xoplaki E., Luterbacher J. Detrended partial-cross-correlation analysis: A new method for analyzing correlations in complex system. Scientific Reports. 2015;5:8143. doi: 10.1103/PhysRevLett.100.084102

12. Lin M., Wang G.J., Xie C., Stanley H.E. Cross-correlations and influence in world gold markets. Physica A: Statistical Mechanics and its Applications.490(15):504-512. doi: https://doi.org/10.1016/j.physa.2017.08.045

13. Qian X.Y., Liu Y.M., Jiang Z.Q., Podobnik B., Zhou W.X., Stanley H.E. Detrended partial cross-correlation analysis of two nonstationary time series influenced by common external forces. Physical Review E. 2015;91(6):062816. doi: 10.1103/PhysRevE.91.062816

---

## [Decision Letter · Decision Letter 1]

26 Mar 2021

Nonlinear Relationship between Money Market Rate and Stock Market Liquidity in China: A Multifractal Analysis

PONE-D-20-32997R1

Dear Dr. Yuan,

We’re pleased to inform you that your manuscript has been judged scientifically suitable for publication and will be formally accepted for publication once it meets all outstanding technical requirements.

Kind regards,

J E. Trinidad Segovia

Academic Editor

PLOS ONE

Additional Editor Comments (optional):

Reviewers' comments:

Reviewer's Responses to Questions

**Comments to the Author**

1. If the authors have adequately addressed your comments raised in a previous round of review and you feel that this manuscript is now acceptable for publication, you may indicate that here to bypass the “Comments to the Author” section, enter your conflict of interest statement in the “Confidential to Editor” section, and submit your "Accept" recommendation.

Reviewer #1: All comments have been addressed

Reviewer #2: All comments have been addressed

Reviewer #3: All comments have been addressed

2. Is the manuscript technically sound, and do the data support the conclusions?

Reviewer #1: Yes

Reviewer #2: Yes

Reviewer #3: Yes

3. Has the statistical analysis been performed appropriately and rigorously? 

Reviewer #1: Yes

Reviewer #2: Yes

Reviewer #3: Yes

4. Have the authors made all data underlying the findings in their manuscript fully available?

Reviewer #1: No

Reviewer #2: No

Reviewer #3: No

5. Is the manuscript presented in an intelligible fashion and written in standard English?

Reviewer #1: Yes

Reviewer #2: Yes

Reviewer #3: Yes

6. Review Comments to the Author

Reviewer #1: This manuscript has been revised according to the reviewers' comments. I am satisfied with this revision therefore I suggest to accept this manuscript.

Reviewer #2: The paper has been thoroughly revised incorporating all my suggestions.The authors have mentioned restrictions in data availability. It is up to the journal authorities in this matter as the reviewer is not aware of the journal policies.

Reviewer #3: (No Response)

7. PLOS authors have the option to publish the peer review history of their article (what does this mean?). If published, this will include your full peer review and any attached files.

Reviewer #1: No

Reviewer #2: No

Reviewer #3: No

---

## [Editor Report · Acceptance letter]

6 Apr 2021

PONE-D-20-32997R1 

Nonlinear Relationship between Money Market Rate and Stock Market Liquidity in China: A Multifractal Analysis 

Dear Dr. Yuan:

I'm pleased to inform you that your manuscript has been deemed suitable for publication in PLOS ONE. Congratulations! Your manuscript is now with our production department. 

Kind regards, 

on behalf of

Dr. J E. Trinidad Segovia 

Academic Editor

PLOS ONE